# Essential Oil of Algerian *Eryngium campestre*: Chemical Variability and Evaluation of Biological Activities

**DOI:** 10.3390/molecules24142575

**Published:** 2019-07-15

**Authors:** Ali Medbouhi, Fethi Benbelaïd, Nassim Djabou, Claire Beaufay, Mourad Bendahou, Joëlle Quetin-Leclercq, Aura Tintaru, Jean Costa, Alain Muselli

**Affiliations:** 1Département de Chimie, Faculté des Sciences, Université de Tlemcen, Laboratoire de Chimie Organique Substances Naturelles et Analyses (COSNA), BP 119, Tlemcen 13000, Algérie; 2Laboratoire Chimie des Produits Naturels (CPN), Université de Corse, UMR CNRS 6134 SPE, Campus Grimaldi, BP 52, 20250 Corte, France; 3Faculté SNV-STU, Université de Tlemcen, Laboratoire LAMAABE, BP 119, Tlemcen 13000, Algeria; 4Pharmacnognosy Research Group, Louvain Drug Research Institute (LDRI), Université Catholique de Louvain, Avenue E. Mounier, 72, B1.7203, B-1200 Bruxelles, Belgium; 5CNRS, Institut de Chimie Radicalaire, Aix Marseille University, UMR 7273, 13397 Marseille, France

**Keywords:** *Eryngium campestre*, essential oil, campestrolide, GC/MS, biological activity

## Abstract

The chemical composition of essential oils extracted from aerial parts of *Eryngium campestre* collected in 37 localities from Western Algeria was characterized using GC-FID and GC/MS analyses. Altogether, 52 components, which accounted for 70.1 to 86.8% of the total composition oils were identified. The main compounds were Germacrene D (0.4–53.4%), Campestrolide (1.6–35.3%), Germacrene B (0.2–21.5%), Myrcene (0.1–8.4%), α-Cadinol (0.2–7.6%), Spathulenol (0.1–7.6%), Eudesma-4(15)-7-dien-1-β-ol (0.1–7.6%) and τ-Cadinol (0.3–5.5%). The chemical compositions of essential oils obtained from separate organs and during the complete vegetative cycle of the plant were also studied. With the uncommon 17-membered ring lactone named Campestrolide as the main component, Algerian *E. campestre* essential oils exhibited a remarkable chemical composition. A study of the chemical variability using statistical analysis allowed the discrimination of two main clusters according to the geographical position of samples. The study contributes to the better understanding of the relationship between the plant and its environment. Moreover, the antimicrobial activity of the essential oil was assessed against twelve strains bacteria and two yeasts involved in foodborne and nosocomial infections using paper disc diffusion and dilution agar assays. The in vitro study demonstrated a strong activity against Gram-positive strains such as *S. aureus*, *B. cereus*, and *E. faecalis*. The cytotoxicity and antiparasitic activities (on *Lmm* and *Tbb*) of the collective essential oil and one sample rich in campestrolide, as well as some enriched fractions or fractions containing other terpenic compounds, were also analyzed. Campestrolide seems to be one compound responsible for the cytotoxic and antileishmanial effect, while myrcene or/and *trans-*β-farnesene have a more selective antitrypanosomal activity.

## 1. Introduction

The genus *Eryngium* L. is probably the most extensive and taxonomically complex genus of Apiaceae family, including about 300 species distributed in temperate regions of all continents, mainly in Eurasia, North Africa, and South America. However, the species richness is unequally distributed between the east and west regions of the globe, around two thirds of species are distributed in America [1].

*Eryngium* species are used as a folk remedy in the treatment of various anti-inflammatory disorders, and used as an emetic, gastrointestinal infusion, an antidote for poisons, as a hypoglycemic agent [2], a remedy for diarrhea, and also as a stimulant, aphrodisiac, antitussive, and diuretic [3], particularly in Turkish folk medicine [4].

*E. campestre* is a perennial plant that measures from 30 to 60 cm in length, with a whitish green, very spiny and thick stalk. The coriaceous basal leaves with a 5–20 cm limb are usually evergreen. The flowers are white with globular or ovoid heads. The fruits are ovate and covered with scales [5]. In Algeria, this species is called “Garrana or Chouk lebiad” and is distributed from the north Sahara to the Mediterranean Sea, in steppes and pastures. No traditional uses are referred to this species by the Algerian population [6].

Previous phytochemical studies on *E. campestre* described the isolation of flavonoids and flavonoacyl derivatives from aerial parts and roots [7,8,9], monoterpene glycosides with cyclohexanone moiety [10], a coumarine derivative [11], and triterpene saponins from the roots [12]. Terpene compounds were also reported from *E. campestre*. The main chemical components of the essential oils obtained from whole aerial parts and/or separated organs, as well as the origins of the studied plants, are tabulated in Table 1. *E. campestre* essential oils were mainly dominated by sesquiterpene compounds. Especially, plants from occidental Mediterranean countries (Portugal, Spain, and Italy) produce principally volatile metabolites that exhibit hydrocarbon sesquiterpenes with germacrane and bisabolane skeletons [13,14,15,16]. In contrast, the plants from the oriental Mediterranean basin (Egypt) seem to produce essential oils characterized by cadinane, guaiane, and oxygenated compounds [17]. The study of the bibliographic data highlights the relative chemical intraspecies variability of *E. campestre* essential oils according to geographical origin of the plants. As suggested by Palá-Paúl et al. [16] the type of soils seems to exert an influence on the chemical composition of the essential oils of this plant.

*E. campestre* has been a subject of many biological tests. The methanol extract of *E. campestre* and the isolated flavonols exhibit moderate to strong antioxidant activity in DPPH radical scavenging and reducing power assays. However the two extracts have no effect on Alzheimer′s disease [9]. Furthermore, both ethanol extracts (from roots and aerial parts) exhibit a strong antioxidant activity [18]. No significant anti-inflammatory and antinociceptive activities have been detected in the ethanolic and aqueous extracts of *E. campestre* [19]. The methanol extract from aerial parts show very strong antitumoral activity on potato tumor cells induced by *Agrobacterium tumefaciens* (ATCC 23341), but no significant antimicrobial activity exhibit against *Staphylococcus epidermidis*, *S. pyogenes*, *Pseudomonas aeruginosa* and *Escherichia coli* [20]. Essential oils prepared from aerial parts were evaluated for their cytotoxic effect on a panel of human cancer cells, namely A375 (human malignant melanoma), MDA-MB 231 cells (human breast adenocarcinoma), and HCT116 cells (human colon carcinoma). Essential oils show a high cytotoxic activity comparable or close to the anticancer drug cisplatin [14]. On the other hand, *E. campestre* essential oil have no inhibitory activity against MRSA strains [21].

The aims of this work was to determine the chemical composition of Algerian *E. campestre* essential oils using Gas Chromatography-Flame Ionization Detector (GC-FID) and Gas Chromatography/Mass Spectrometry (GC/MS), and to study the biological activities of the essential oils against fourteen microorganisms involved in foodborne and nosocomial infections using paper disc diffusion and dilution agar methods and the cytotoxicity and antiparasitic activities on WI38 and J774 cells and on *Trypanosoma brucei brucei* (*Tbb*) and *Leishmania mexicana mexicana* (*Lmm*). The study highlights the contribution of chemical data to shed light on the ecology of the species. For this purpose, the chemical variability of the volatile components was studied to identify possible correlations between environmental factors and essential oil compositions of 37 samples. Our study focuses on the originality of the chemical composition of Algerian *E. campestre*. The chemical compositions of *E. campestre* essential oils obtained from separated organs and during the complete vegetative stage of the plant were also investigated.

## 2. Results

### 2.1. Harvest Locations

The aerial parts of E. campestre were harvested from May to July 2016 from 37 localities (S1–S37) of Tlemcen (Northwest of Algeria). The sample numbers and information concerning the environmental parameters of the harvest locations are listed in Figure 1 and Table 2. The plant material was harvested from high steppe plains characterized by low organic matter and low water resources (Area 1: S1–S16) and from the mountains of Traras and the plains of Tlemcen rich with water accumulations (Area 2: S17–S37). The two areas are separated by a mountain range called Traras. Hydrodistillation from the fresh aerial parts of the 37 Algerian samples yielding (*w*/*w*) 0.1–0.2% of yellow essential oils with musk scent.

### 2.2. Chemical Composition of E. campestre Essential Oils

The individual oil samples were pooled to produce *E. campestre* collective oil (EC-CO). The EC-CO was fractionated with column chromatography using a gradient of polarity with hexane and diisopropylether and followed by UV detection to give two fractions, polar (3500 mg) and apolar (900 mg). The polar fraction was fractionated again to produce 87 sub-fractions. The chemical composition of EC-CO was established by analyzing all the fractions and sub-fractions obtained by successive chromatography columns by GC-FID and GC/MS. The detailed chemical compositions of the 37 sample oils can be obtained as Appendix A from the authors. In total, seven monoterpenes, 38 sesquiterpenes, and seven non-terpenic compounds were identified (Table 3). The chemical composition of *E. campestre* collective oil (EC-CO) was dominated by sesquiterpene compounds (60.5%), among them oxygenated and hydrocarbon compounds amounted for 27.4 and 33.1%, respectively.

The main components (Figure 2) were Germacrene D (26, 15.2%), Campestrolide (52, 10.3%), Spathulenol (39, 4.8%) and α-cadinol (46, 5.5%). Campestrolide 52 was an uncommon and naturally found 17-membered ring lactone previously isolated from the hexane extract of Algerian *E. campestre* [22]. To the best of our knowledge, our study reports, for the first time, this 17-membered ring featuring conjugated acetylenic bonds as the main component of *E. campestre* essential oil.

In addition, the essential oils of separate organs from the location S1 (Lalla Setti) were investigated for their chemical compositions. As seen in Table 3, whatever the studied organ, *E. campestre* yielded sesquiterpene-rich essential oils, with contents superior to 59.8% for all organs. Among them, hydrocarbon sesquiterpenes were the main components, they amounted for 56.5%, 45.3% and 43.5% in the essential oils obtained from stems, flowers and leaves and higher contents were found in the essential oil prepared from roots (60.3%). Germacrene D was the main compound in all organs (29.1–38.3%). The other main components were (i) (E)-β-Farnesene (26.0%) in the roots oil while their amounts were minor in the three other organs (1.4–9.8%), (ii) Campestrolide amounted for 16.4% and 14.6% in the leaf and stem essential oils vs. 9.5% and 4.5%, respectively, in the root and flower essential oils; and (iii) α-Bisabolol which the relative amount reaches to 8.0% in the flowers essential oil vs. 1.8–3.9% in the other organs.

To gain more insights about the plant ecology, the variation in the relative amounts of the oil-components from the location S1 was tracked during one plant-vegetative cycle (May to July) and the results are reported in Table 3. During the first stage of the vegetative cycle (May to early June), the plant exhibited stems and leaves only and the corresponding essential oil was dominated by oxygenated compounds (64.1%). Campestrolide was the main component (50.9%) and the oxygenated sesquiterpenes amounted for 12.0%. In this stage of vegetative growth, the production of volatile hydrocarbons (20.3%) by the plant was weaker and Germacrene D amounted for 11.2%. During the progress of the plant-vegetative cycle, the sharp decrease of Campestrolide amount (32.1 to 8.1%) matched the increase of Germacrene D (19.5–30.4%). Therefore, at the end of the studied period, i.e., full bloom (June and July), the essential oil produced by the plant exhibited a higher amount of hydrocarbon sesquiterpenes (65.9%) than at the start of the plant-vegetative cycle (31.7%). It should be noted that the amount of leaves during the early vegetative stages is greater than after the flowering stage and as described above, leaves are the organs that exhibited the highest concentration of Campestrolide (16.3%). Therefore, it can be asserted that the proportion of leaves affected the essential-oil composition of the whole plant.

Relative to the chemical composition of *E. campestre* essential oils from different origins reported in the literature [13,14,15,16,17], the Algerian essential oil EC-CO displayed a remarkable originality. As the plant from occidental Mediterranean countries such as Italy, Spain, and Portugal yielded essential oils dominated by hydrocarbon compounds, Algerian *E. campestre* yielded essential oil characterized by Germacrene D and Campestrolide. To the best of our knowledge, Campestrolide was never detected in *Eryngium* essential oils. Hence, this chemical, particularly, might be used as a discriminant marker for the taxonomic study.

### 2.3. Chemical Varaibility of Algerian E. campestre Essential Oils

In order to highlight a possible chemical variability, the chemical composition of the 37 essential oil samples isolated from the aerial parts of Algerian *E. campestre* samples harvested at the flowering was compared using statistical analysis. A principal component analysis (PCA, Figure 3a) was constructed using axes 1 and 2 that accounted for 81.13% of the variance. The first dimension (Dim 1) was positively correlated with oxygenated compounds (sesquiterpenes and non terpenic) and negatively correlated with hydrocarbon compounds (monoterpenes and sesquiterpenes). The plot suggests that there are two main groups (PCA, Figure 3a) in accordance with the general structure of the dendrogram obtained from the Cluster Analysis (CA, Figure 3b): Group I includes 16 samples from locations S1–S16, and group II includes 21 samples (S17–S37). Sample oils of group I were characterized by a higher amount of oxygenated sesquiterpenes (6.3–33%) and non terpenic oxygenated compounds (18.7–37.4%) than those of group II (1.7–13.2%) and (6.5–28%) respectively.

The statistical distribution of the oil samples focused on the correlation between the chemical composition of individual oil samples and the geographical distribution. Group I regrouped all specimens collected in area 1, and group II included all specimens harvested in area 2. The sample oils from Group I were prepared from plants harvested in the south of the mountains of Tlemcen (Figure 1). This zone is the high steppe plains localized between the Tellian Atlas in the north and the Saharan Atlas in the south. It is characterized by bioclimatic stages which extend from the fresh semi-arid to the fresh semi-wet. Steppe soils are characterized by the presence of limestone accumulation, low organic matter content, and low water resources [23]. The sample oils from Group II were prepared from plants located in the area bounded by the mounts of Tlemcen to the south and the littorals to the north. In this zone, we find a semi-arid climate from soft to hot. The soils in this zone are calcareous humiferous and rich in organic matter. These soils are well watered by yearly precipitations, deep water tables, and watershed, which promote annual crop yields [24]. It is probable that all these environmental parameters exert an influence on chemical composition of *E. campestre* essentials oils. As the genetic diversity of *E. campestre* from Central Europe was reported, it could be interesting to lead a genetic study of Algerian species in order to examine the influence of the genetic patrimony on the chemical variability of the plant-volatiles.

### 2.4. Growth Inhibition of Bacteria by Essential Oil

The results of susceptibility of Algerian *E. campestre* collective oil testing with the agar diffusion method and minimum inhibitory concentrations (MIC) are presented in Table 4. The qualitative agar diffusion method is commonly used as a preliminary method prior to more detailed studies. Thus, MIC assays are generally more sensitive and allow the quantitative determination of essential oils’ antibacterial efficiency. In the case of Gram-positive reference strains, inhibition zone values for Gentamicine were 10 to 21 mm. Our results showed that *E. campestre* essential oil has a very interesting activity against *Staphylococcus aureus* ATCC 25923, ATCC 33862 and ATCC 29213, *Bacillus cereus* ATCC 11778 and *Enterococcus faecalis* ATCC29212, where the largest diameters of inhibition ranging from 20 to 35 mm were recorded. MIC assays revealed the sensitivity of the same strains to *E. campestre* essential oil. Both other Gram-positive reference strains, *Listeria monocytogenes* ATCC19115 and *Bacillus subtilis* ATCC6633, were resistant to *E. campestre* essential oil. Gentamicine inhibition zone varied from 11 to 20 mm for Gram-negative reference strains. With inhibition diameters equal to 6 mm, *E. campestre* essential oil was not active against the five Gram-negative strains tested. In the same way, *E. campestre* essential oil showed a very low activity against yeasts, since inhibition diameters did not exceed 6 mm (vs. 20–22 mm using Amphotericin B as antibiotic) and MIC was not measured.

Our results are in accordance with the well-known concept indicating that due to the structural differences of their external membranes, Gram-negative are supposed to be more resistant than Gram-positive strains. In Gram-negative bacteria, the lipopolysaccharide-rich outer membrane constitutes an effective barrier of permeability whose negative surface charges prevent the diffusion of hydrophobic molecules [25]. *E. campestre* essential oil was active against five of the seven Gram-positive strains tested, while it was inactive against the five Gram-negative strains investigated. The antibacterial activity of *E. campestre* essential oil could be directly connected to the essential oil chemical composition. Since hydrocarbon compounds are known to be weak actives against bacteria, essential oils dominated by oxygenated components such as, oxides and alcohols possessing the strongest antibacterial properties [25]. Hence, the activity of *E. campestre* essential oil could be conditioned to the presence of their main components, such as Germacrene D, Spathulenol and α- and τ-Cadinol which exhibited moderate or strong antimicrobial activity [26,27]. However, the antimicrobial activity of essential oils cannot be exclusively related to their main compounds. Indeed, the minor components, such as Myrcene, Limonene, Caryophyllene oxide, and Ledol provide more activity of EC-CO against microorganisms by interactions and phenomena of synergy with the main compounds [28]. Finally, the antibacterial activity of Campestrolide has never been described in the literature and could be further investigated.

### 2.5. Cytotoxity and Antiprotozoal Activities of E. campestre Essential Oil

Given the presence of campestrolide and its relatively selective antitrypanosomal effect [22], we also analyzed the activity of *E. campestre* collective EO (EC-CO) on *Trypanosoma brucei brucei* (Tbb) and *Leishmania mexicana mexicana* (Lmm), as well as its cytotoxicity on WI38 and J774 cells. To evaluate the link with the campestrolide content, we also tested S31 EO sample (containing 30% campestrolide) as well as two fractions obtained after column chromatography. One is enriched in campestrolide (EC1 containing about 70% campestrolide) and one containing (ECC1) only 1.2% campestrolide but rich in myrcene (38%) and trans-farnesene (9%). The results are given in Table 5.

The results show that campestrolide seems to be responsible, at least partially, for the cytotoxicity and antileishmanial activity of the EOs and enriched fractions. This compound is also responsible for a part of the antitrypanosomal activity but myrcene or/and *trans-*β-farnesene are also effective on *Trypanosoma* with an improved selectivity. Myrcene was already shown to possess antitrypanosomal activity (IC50: 2.24 ± 0.25 μg/mL [29]), hence, given the activity of the mixture with *trans-*β-farnesene, it may be concluded that *trans-*β-farnesene is more active or that synergy occurs.

## 3. Materials and Methods

### 3.1. Plant Material

The fresh aerial parts (including stems, leaves, and flower heads) of *E. campestre* were harvested from May to July 2016 from 37 localities of Tlemcen, in the west of Algeria (Samples S1–S37). In Lalla Setti (S1), also the roots were collected, and the stems, leaves, and flower heads were separated from the total aerial parts to study the essential oils obtained from the separate organs. For the study of seasonal variations, the fresh aerial parts were harvested every 20 days (May to July 2016) from plants growing in Lalla Setti (S1). The botanical identification was performed by the botanic department of the University of Tlemcen (Algeria), and voucher specimens were deposited in the same institution. The sample numbers, geographical origin of the different samples, and the voucher code of each specimen analyzed are listed in Figure 1.

### 3.2. Essential-Oil Isolation

For each sample, the fresh plant material (200–400 g) was subjected to hydrodistillation (5 h) using a Clevenger-type apparatus according to the method recommended in the European Pharmacopoeia [30]. A seen in Table 3. the essential oil yields were expressed in % (*w*/*w*), based on the weight of the fresh plant material.

### 3.3. Oil Isolation

The *E. campestre* collective essential oil (EC-CO) was obtained by the pooling of the 37 individual essential oils. A portion of 5 g of EC-CO was submitted to column chromatography (CC) on a silica gel column (200–500 mm, 12 g, Clarisep^®^, Bonna Agela Technologies, Willington, USA) with a Combi Flash apparatus (Teledyne ISCO, Lincoln, USA) equipped with a fraction collector monitored by an UV detector. A nonpolar fraction (Hydrocarbon Fraction: HF; 1024 mg) and a polar fraction (Oxygenated Fraction: OF; 3590 mg) were obtained by elution with hexane and diisopropyl ether (DIPE). The OF was further divided into eighty-seven polar sub-fractions obtained by elution with a hexane-DIPE gradient solvent system with increasing proportions of DIPE. All fractions were analyzed by GC and GC-MS.

### 3.4. GC-FID Analysis

Analyses were carried out using a Perkin-Elmer Autosystem XL GC apparatus (Walthon, MA, USA), equipped with a dual flame ionization detection (FID) system and fused-silica capillary columns, namely, Rtx-1 (polydimethylsiloxane) and Rtx-wax (polyethyleneglycol) (60 m x 0.22 mm i.d.; of a film thickness of 0.25 μm). The oven temperature was programmed from 60 to 230 °C at 2 °C/min and then held isothermally at 230 °C for 35 min, and helium was employed as the carrier gas (1 mL/min). The injector and the detector temperatures were maintained at 280 °C and the samples were injected (0.2 μL of pure oil) in the split mode (1:50). The RI of each compound was determined relative to the retention times of a series of n-alkanes (C5–C30) by linear interpolation, using the Van den Dool and Kratz (1963) equation with the aid of the software from Perkin–Elmer (Total Chrom navigator, 6.3.1, Shelton, CT, USA). The relative percentages of the extract constituents were calculated from the GC peak areas by the normalization procedure, without the application of correction factors.

### 3.5. GC-MS Analysis

The essential oils and the fractions obtained by CC were investigated using a Perkin–Elmer TurboMass quadrupole detector directly coupled to a Perkin–Elmer AutoSystemXL (Walton, MA, USA) equipped with the two same fused-silica capillary columns as described above. Both columns were used with the same MS detector. The analyses were consecutively carried out on the nonpolar and on the polar column. Hence, for each sample, two reconstructed ionic chromatograms (RIC) were provided, which were investigated consecutively. The GC conditions were the same as described above, and the MS parameters were as follows: Ion-source temp., 150°; ionization energy; 70 eV; EI-MS spectra acquired over a mass range of 35–350 Da during a scan time of 1 s. The injection volumes for the oil and the fractions were 0.1 and 0.2 µL, resp.

### 3.6. Compound Identification and Quantification

The identification of individual components was based (i) on the comparison of the retention indices (RIs) determined on the polar and nonpolar columns with those of authentic compounds or literature data [31,32], (ii) on computer matching of the mass spectra with commercial mass-spectral libraries and on the comparison of the mass spectra with those listed in our homemade library built of mass spectra of authentic compounds or literature data [33]. Quantification of the oil components was performed using the methodology adapted in our laboratory [34]. Briefly, the compound quantification was carried out using peak normalization, including FID response factors (RFs) rel. to tridecane (0.7 g/100 g) used as internal standard and expressed as normalized % abundances.

### 3.7. Statistical Analysis

Data analyses were performed using principal component analysis (PCA) and cluster analysis (CA). Both methods aim at reducing the multivariate space in which objects (oil samples) are distributed, but are complementary in their ability to present results [35]. Indeed, PCA provides the data for plots in which both objects (oil samples) and variables (chemicals class) are represented, while canonical analysis informs a classification tree in which objects (sample locations) are gathered. The PCA was carried out using the function PCA of the statistical R software (R Foundation, Institute for Statistics and Mathematics, Austria). The discriminate variables (volatile components) have been selected using a function of the statistical software. A dendrogram (tree) was produced by canonical analysis using Ward′s method of hierarchical clustering based on Euclidean distances between pairs of oil samples.

### 3.8. Microbial Strains

Fourteen reference strains were used, including two yeasts: Candida albicans ATCC 10231 and Candida albicans IPP444. Seven Gram-positive bacteria: Enterococcus faecalis ATCC 29212, Staphylococcus aureus ATCC 25923, Staphylococcus aureus ATCC 33862, Staphylococcus aureus ATCC 29213, Bacillus subtilis ATCC 6633, Bacillus cereus ATCC 11778, and Listeria monocytogenes ATCC 19115 and five Gram-negative bacteria: Escherichia coli ATCC 25922, Klebsiella pneumoniae ATCC 70603, Salmonella enteritidis ATCC 2453, Pseudomonas aeruginosa ATCC 27853, and Pseudomonas fluorescens ATCC 13525.

### 3.9. Preparation of Inoculum

The pure strains were inoculated into tubes containing 5 mL of Brain Heart Infusion (CondaPronadisa™, Madrid, Spain) were then incubated at 37 °C for 18 h. A concentration of 0.5 McFarland has been adjusted corresponding to 10^7^ CFU/mL, according to the Clinical and Laboratory Standards Institute CLSI [36].

### 3.10. Sensitivity Tests—Disc Diffusion Method

The susceptibility of strains to the essential oil was tested according to the method described by Bauer and Kirby [37]. A culture of 18–24 h of the tested strains on Müller–Hinton broth was prepared at a concentration of 10^7^ CFU/mL and then seeded on Müller-Hinton agar (Fluka Bio Chemika, Spain) and Sabouraud (FlukaBioChemika, Spain) for the yeast *Candida albicans* by swab as recommended by CLSI [36]. A sterile disc of 6 mm diameter impregnated with 5 µL of essential oil was aseptically deposited in the center of the inoculated plate.

### 3.11. MICs

The minimum inhibitory concentration (MIC) was performed by the method of microplate (96 wells) round bottom using as reported in literature [38]. The essential oil underwent successive dilution of ½ in the outer tubes in Müller–Hinton broth (FlukaBioChemika, Spain) by adding Tween 80 to a concentration of 1% (*v*/*v*) to obtain a homogeneous solution of essential oil and broth. A second solution was prepared containing only Müller–Hinton broth and Tween 80 at a concentration of 1%, this concentration was used as a complement of dilutions with broth and Tween 80 for grader the same concentration of Tween 80; 1% for each concentration of essential oil. Inocula of 10^7^ CFU/mL were diluted by 1/100 for a concentration of 10^5^ CFU/mL. On a microplate, 180 µL of bacterial suspension 10^5^ CFU/mL were deposited inside the wells. Then 20 µL of the solution of the essential oil was added. The final concentration of Tween 80 is 0.1% (*v*/*v*) to each well and the final concentrations of the essential oil ranged from 0.0078 to 4%.

### 3.12. Cytotoxity and Antiprotozoal Activities

The EOs, fractions and pure campestrolide (91.9%), were evaluated in vitro for their cytotoxicity on two mammalian cell lines: Cancerous macrophage-like murine cells (J774) and non-cancer human fibroblasts (WI38). Their antileishmanial activities were investigated on *Leishmania mexicana mexicana* (Lmm MHOM/BZ/84/BEL46) promastigotes and *Trypanosoma brucei brucei* (Tbb strain 427) bloodstream forms. The tests were performed as described in [39,40] using stock solutions prepared in DMSO at 10 mg/mL. For the cytotoxicity assay, the samples were tested in eight serials 1.7-fold dilutions (150 μL transferred into 100 μL fresh medium) in the 96-well microtiter plates (concentration range: 0.7–25 μg/mL). Camptothecin (Sigma-Aldrich, St.-Louis, MO, USA) was used as a positive control with 5-fold dilutions (concentration range: 0.00032–25 μg/mL) from a 10 mg/mL DMSO stock solution. For the antiparasitic assays, the samples were tested in eight serial 2-fold dilutions (concentration range: 0.20–25 or 0.10–12 μg/mL on Leishmania and Trypanosoma, respectively). Suramine sodium and pentamidine isethionate salts (Sigma–Aldrich, St.-Louis, MO, USA) were used as positive controls with 3-fold dilutions (0.0046–10 μg/mL) from stock solutions of 2 mg/mL. A maximum of 0.5% DMSO was previously verified to be non-toxic in all the biological assays. The IC50 values were determined using Microsoft Excel and GraphPad Prism 7.0 software (GraphPad, San Diego, CA, USA), based on a nonlinear regression. The selectivity index was calculated in comparison to the WI38 cytotoxicity to assess the therapeutic potentialities as antiparasitic hit.

## 4. Conclusions

The present study investigated, for the first time, the chemical composition of Algerian *E. campestre* essential oils isolated from different plant organs as well as the seasonal variation of the essential oil. These results provided a better insight on the secondary-metabolite production occurring during the plant-life. The occurrence of campestrolide, a 17-membered lactone, provides a remarkable originality to the Algerian *E. campestre* essential oil comparing to previous chemical data of *Eryngium* essential oils reported in the literature. Statistical analyses highlighted that two chemical compositions of the Western Algerian *E. campestre* essential oils were distinguished. The ecological relevance of these results was supported by the correlation between the chemical composition and environmental parameters (geographical position, bioclimate and nature of soils) of individual oil samples. In addition, the antimicrobial activity of collective essential oil against various bacterial strains and yeasts was evaluated for the first time. *E. campestre* essential oil from Algeria were most active against *S. aureus*, *B. cereus,* and *E. faecalis*. Thus, the essential oil could be considered as a good way to investigate in order to struggle against foodborne and toxi-infectious pathogens. We also analyzed the cytotoxicity and antiparasitic activities (on *Lmm* and *Tbb*) of the collective EO and one sample rich in campestrolide, as well as some enriched fractions or fractions containing other terpenic compounds and found that campestrolide seems mainly responsible for the cytotoxic and antileishmanial effect, while myrcene or/and *trans-*β-farnesene have a more selective antitrypanosomal activity. Finally, the present results might be useful for the future commercial valorization of this species by producing *E. campestre* essential oil.

## Figures and Tables

**Figure 1 molecules-24-02575-f001:**
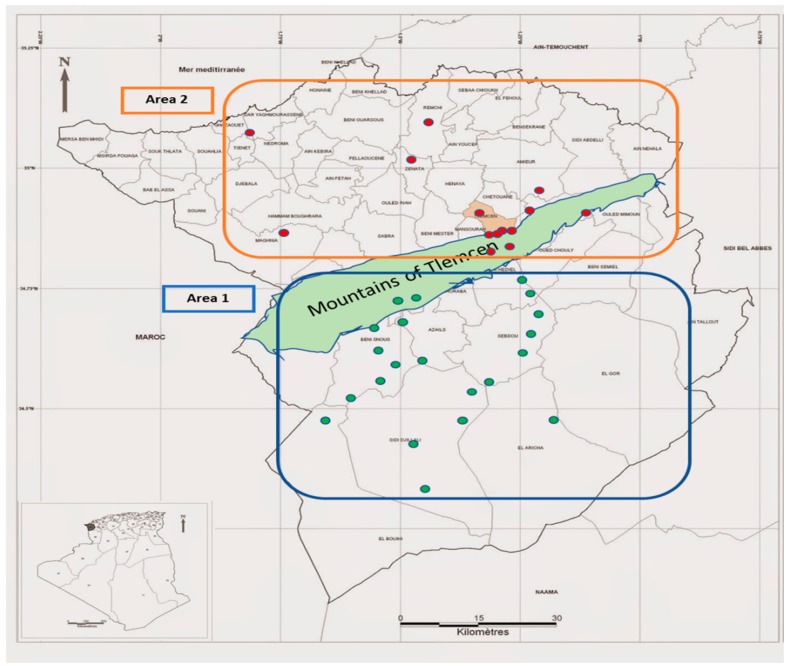
Geographical localization of the Algerian *Eryngium campestre* samples (see Table 2 for more details).

**Figure 2 molecules-24-02575-f002:**
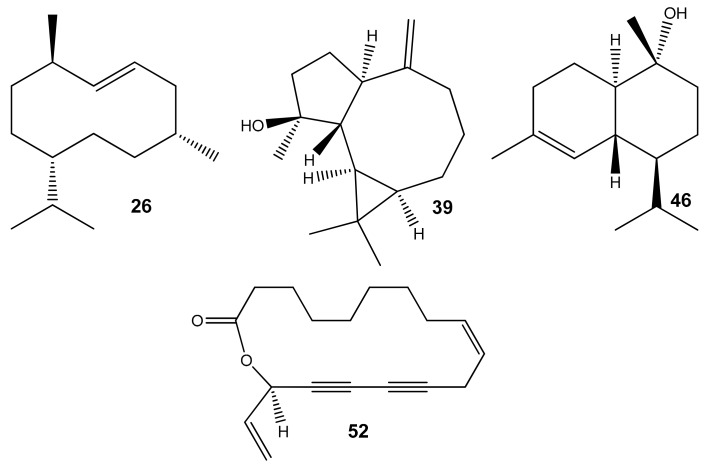
Main components of *E. campestre* essential oil from Algeria.

**Figure 3 molecules-24-02575-f003:**
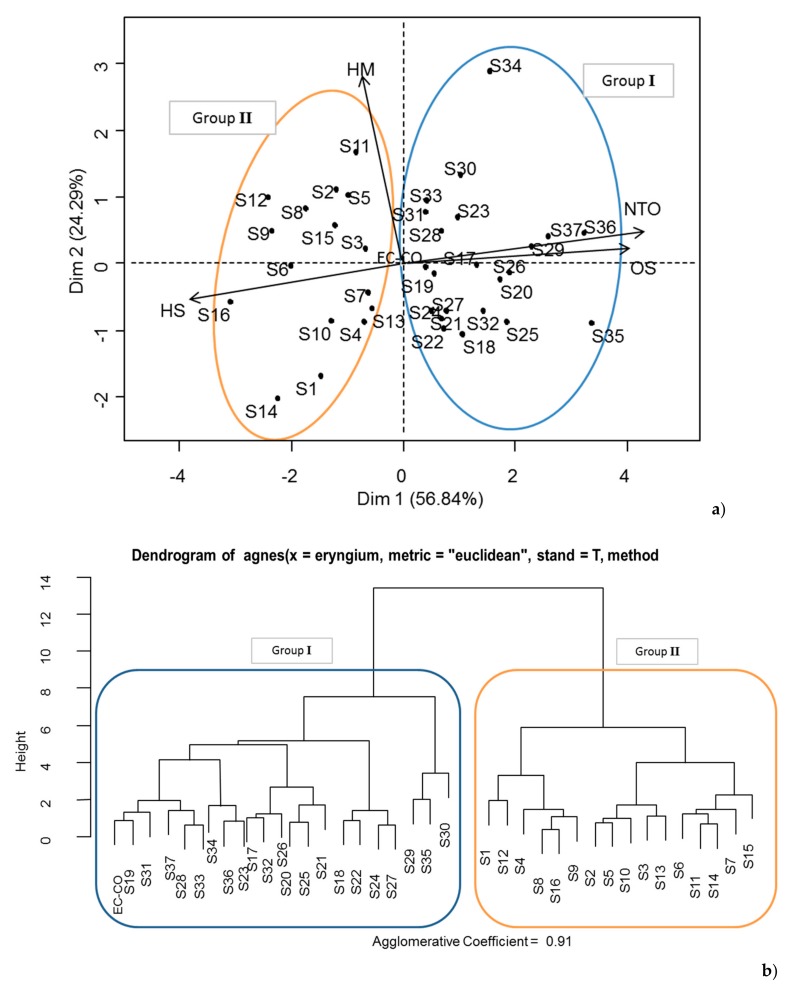
(**a**) Individual factors map obtained by principal component analysis performed on the oxygenated and hydrocarbon components of the 37 Algerian *Eryngium campestre* essential oil samples. Oxygenated compounds: (OS: Oxygenated Sesquiterpenes and NTO: Non Terpenic Oxygenated); Hydrocarbons compounds: (HM: Hydrocarbonated Monoterpenes and HS: Hydrocarbonated Sesquiterpenes). For more details on Samples S1–S37, cf. Appendix A. (**b**) Dendrogram obtained using cluster analysis of chemical compositions of *E. campestre* essential oils from Algeria.

**Table 1 molecules-24-02575-t001:** Main components identified in *E. campestre* essential oils.

Origins	Plant Organs	Main Components	%	Ref.
Portugal	Aerial parts	(*E*)-β-Farnesene β-Bisabolene	31.31 7.8	[13]
		Germacrene D Germacrene A β-Elemene	18.4 9.2 19.7	
		β-Bisabolene Germacrene D α-Cadinol	15.6 12.3 8.2	
Italy	Aerial parts	Germacrene D Allo-Aromadendrene Spathulenol Ledol	13.8 7.7 7.0 5.7	[14]
Italy	Inflorescences	Germacrene D Apiol α-Himachalene	49.5 19.5 15.1	[15]
Spain	Inflorescences	Germacrene D β-Curcumene Myrcene (*E*)-β-Farnesene	30.3–40.3 0.7–22.2 3–21.7 0.1–19	[16]
Egypt	Aerial parts	γ-Cadinen-15-al Spathulenol Octanoic acid α-Curcumene	23.3 10.7 9.8 8.6	[17]

**Table 2 molecules-24-02575-t002:** Sample numbers, geographical details, and voucher codes of Algerian *Eryngium campestre* samples S1–S37.

N°	Sample Location of Harvest	GPS		Alt	Voucher Codes	Area
		Latitude	Longitude			
S1	Lalla setti	34°51′52.42″ N	1°18′18.09″ O	996	EC01	Area 1: high steppe plains/Limestone acumulation soils/Low organic matter/Law water resources
S2		34°51′52.68″ N	1°18′20.72″ O	996	EC02	
S3		34°51′52.35″ N	1°18′20.72″ O	997	EC03	
S4		34°51′38.24″ N	1°19′4.80″ O	1037	EC04	
S5	Bouhanak	34°52′38.75″ N	1°21′33.88″ O	774	EC28	
S6	Mafrouch	34°50′55.59″ N	1°17′53.21″ O	1148	EC29	
S7		34°49′45.37″ N	1°18′44.75″ O	1133	EC30	
S8	Zenata	34°56′51.27″ N	1°26′58.89″ O	300	EC33	
S9	Remchi	35°03′00″ N	1° 26′ 00″ O	82	EC34	
S10	Maghniya	34° 51′ 42″ N	1° 43′ 50″ O	455	EC36	
S11	Chlaida	34°57′8.14″ N	1°13′28.33″ O	609	EC37	
S12	Tirni	34°49′1.90″ N	1°19′36.26″ O	1159	EC05	
S13		34°47′8.00″ N	1°20′37.88″ O	1213	EC06	
S14	Bni aad	35° 2′32.70″ N	1°40′9.43″ O	228	EC35	
S15	Ain lekbira	35° 2′32.70″ N	1°40′9.43″ O	228	EC31	
S16	Mdig	34°58′6.66″ N	1°15′48.65″ O	497	EC32	
S17	Sebdou	34°40′55.15″ N	1°18′47.37″ O	1013	EC07	Area 2: Mountains of Traras and palins of Tlemcen/calcareous humiferous soils/Rich in organic matter/Rich in water resources
S18		34°39′43.52″ N	1°19′21.42″ O	920	EC08	
S19		34°38′15.18″ N	1°20′9.95″ O	938	EC09	
S20	Abed	34°28′42.76″ N	1°40′5.82″ O	1446	EC18	
S21	Bni behdel	34°41′10.07″ N	1°30′16.82″ O	780	EC10	
S22		34°42′38.22″ N	1°27′12.97″ O	776	EC11	
S23	Sidi bounoir	34°42′26.39″ N	1°19′9.80″ O	1110	EC12	
S24		34°40′55.15″ N	1°18′47.37″ O	1030	EC13	
S25	Sebdou/Sid djillali	34°36′0.53″ N	1°23′35.57″ O	997	EC14	
S26		34°34′26.89″ N	1°26′6.66″ O	1005	EC15	
S27	Abed/Lekhmis	34°37′5.86″ N	1°35′34.98″ O	867	EC20	
S28		34°34′40.59″ N	1°36′4.15″ O	980	EC21	
S29		34°33′28.61″ N	1°37′21.11″ O	1123	EC22	
S30		34°32′22.19″ N	1°38′34.81″ O	1294	EC23	
S31		34°31′18.90″ N	1°40′55.59″ O	1360	EC24	
S32		34°29′19.73″ N	1°40′55.63″ O	1253	EC25	
S33	Boughadou	34°30′49.71″ N	1°30′20.98″ O	1336	EC19	
S34	Sid djilali	34°26′11.75″ N	1°34′31.83″ O	1245	EC16	
S35		34°23′27.94″ N	1°36′23.71″ O	1163	EC17	
S36	Laouadj	34°29′3.67″ N	1°16′18.01″ O	1009	EC27	
S37	Lekhmis	34°38′20.61″ N	1°33′45.40″ O	840	EC26	

**Table 3 molecules-24-02575-t003:** Chemical composition of *Eryngium campestre* collective essential oil (EC-CO); —of essential-oils isolated from the aerial parts during the plant vegetative cycle (May–July) and —of essentials oils isolated from the different organs (flowers, leaves, stems, and roots) of sample S1.

No ^a^	Components	*l*RI_a_^b^	RI_a_ ^c^	RI_p_ ^d^	EC-CO ^e^	Seasonal Variation ^f^				Plant Organs ^g^				Identification ^h^
						04 May	24 May	15 June	05 July	Roots	Stems	Leaves	Flowers	
1	β-Pinene	970	972	1110	0.1	0.1	0.1	0.1	0.3	tr	tr	tr	tr	RI, MS
2	Myrcene	979	982	1153	2.7	0.3	2.4	4.6	8.4	3.2	1.9	1.4	4.7	RI, MS
3	p-Cymene	1011	1013	1258	tr	tr	tr	tr	tr	tr	tr	tr	tr	RI, MS
4	Limonene	1020	1022	1199	0.1	0.1	0.1	0.1	tr	tr	tr	tr	0.2	RI, MS
5	(*Z*)-β-Ocimene	1024	1026	1230	0.1	0.1	tr	tr	tr	0.1	tr	tr	tr	RI, MS
6	δ-Terpinene	1047	1049	1243	tr	tr	0.3	0.2	0.1	0.2	tr	tr	tr	RI, MS
7	Nonan-2-one	1070	1076	1388	tr	0.2	tr	0.3	0.2	tr	tr	tr	tr	RI, MS
8	Terpinolene	1078	1079	1280	tr	tr	0.2	0.1	0.2	tr	tr	tr	tr	RI, MS
9	Nonanal	1083	1082	1394	0.1	0.1	0.1	tr	0.1	0.1	tr	tr	0.1	RI, MS
10	Decanal	1185	1184	1498	0.3	0.3	0.2	tr	0.2	0.2	tr	0.2	tr	RI, MS
11	(*E*)-2-Decenal	1248	1247	1652	0.4	0.4	0.3	0.1	0.3	0.1	tr	tr	0.2	RI, MS
12	(*E*)-2-Undecanal	1343	1348	1726	tr	tr	tr	tr	0.1	tr	tr	tr	tr	RI, MS
13	α-Copaene	1379	1375	1438	0.9	0.9	0.6	0.2	1.1	0.3	0.6	0.4	tr	RI, MS
14	β-Bourbonene	1385	1383	1515	0.1	tr	0.2	tr	0.1	tr	0.2	0.1	0.6	RI, MS
15	β-Elemene	1388	1387	1589	3.0	0.4	3.7	4	1.5	0.4	0.9	2.5	0.1	RI, MS
16	β-Ylangene	1420	1417	1562	0.6	0.1	0.7	2.1	0.8	0.5	0.7	0.7	tr	RI, MS
17	(*E*)-β-Caryophyllene	1424	1426	1591	tr	0.1	tr	tr	0.3	tr	tr	0.3	0.1	RI, MS
18	δ-Elemene	1431	1432	1581	1.2	0.2	1.4	0.8	0.7	tr	0.7	tr	0.1	RI, MS
19	trans-α-Bergamotene	1432	1433	1575	1.2	tr	0.2	0.4	0.2	tr	tr	tr	tr	RI, MS
20	(*E*)-β-Farnesene	1448	1449	1661	0.4	0.9	2.1	5.1	8.2	26.0	9.8	1.4	6.2	RI, MS
21	Alloaromadendrene	1451	1454	1631	0.2	0.1	0.2	tr	0.2	tr	tr	tr	tr	RI, MS
22	α-Humulene	1456	1457	1665	0.3	0.1	tr	tr	0.1	tr	0.2	0.4	tr	RI, MS
23	4,5-di-epi-Aristolochene	1467	1465	1665	0.3	1	0.3	tr	tr	0.1	tr	tr	tr	RI, MS
24	δ-Muurolene	1467	1469	1683	0.9	0.1	tr	1	tr	tr	tr	tr	tr	RI, MS
25	α-Curcumene	1470	1471	1742	1.3	0.3	0.9	tr	0.5	tr	0.8	0.4	0.1	RI, MS
26	Germacrene D	1476	1480	1704	15.2	11.9	19.5	28.9	30.4	29.1	38.8	33.7	37.0	RI, MS
27	β-Selinene	1483	1484	1712	0.9	0.8	0.4	0.3	0.4	tr	0.2	0.1	0.1	RI, MS
28	α-Muurolene	1496	1503	1720	0.8	0.3	0.5	0.8	1.9	2	1.8	1.8	0.2	RI, MS
29	β-Bisabolene	1500	1500	1720	0.2	0.2	0.2	0.4	1.4	0.8	0.3	tr	0.2	RI, MS
30	Sesquicineole	1505	1506	1737	0.5	0.4	0.1	0.2	0.2	1.1	0.2	0.5	0.1	RI, MS
31	δ-cadinene	1516	1513	1752	0.3	0.1	tr	tr	0.2	tr	tr	tr	tr	RI, MS
32	β-curcumene	1509	1510	1733	0.2	0.5	0.1	0.9	0.3	0.2	tr	0.3	tr	RI, MS
33	δ-cadinene	1516	1514	1752	1.5	0.2	tr	0.3	1.5	tr	0.2	0.3	tr	RI, MS
34	(*E*)-α-bisabolene	1532	1531	1753	0.5	0.7	1.5	tr	tr	0.6	0.8	0.9	0.5	RI, MS
35	β-Elemol	1535	1534	2072	0.8	tr	tr	0.3	0.1	tr	0.2	tr	tr	RI, MS
36	7-epi-trans-Sesquisabinene hydrate	1543	1547	1991	1.0	0.2	2.5	3.3	tr	0.3	0.1	0.2	tr	RI, MS
37	Salvial-4(14)-ene-1,5-epoxide	1545	1548	1941	2.1	0.9	tr	0.2	0.2	0.6	tr	0.8	0.6	RI, MS
38	Germacrene B	1553	1551	1827	3.1	0.8	0.3	0.2	0.4	0.3	0.5	0.2	0.1	RI, MS
**39**	Spathulenol	1563	1562	2103	4.8	0.9	0.7	9.6	6.3	0.6	2.3	0.5	1.4	RI, MS
40	Caryophyllene oxide	1576	1570	1980	0.2	3.1	1.1	0.8	2.6	0.2	0.3	0.7	1.2	RI, MS
41	Salvial-4(14)-en-1-one	1583	1577	2005	1.8	1.7	0.4	0.5	1	0.8	1.9	4.2	4.5	RI, MS
42	Ledol	1600	1602	2030	1.6	0.8	0.4	0.4	0.5	0.2	0.4	0.5	0.6	RI, MS
43	1,10-di-epi-Cubenol	1610	1611	2054	1.4	tr	1.8	2.2	0.1	0.5	0.2	1.4	0.3	RI, MS
44	Caryophylla-4(14),8(15)-dien-5-α-ol	1626	1624	2285	0.3	0.2	2.3	2	0.5	0.3	tr	0.5	1.9	RI, MS
45	τ-Cadinol	1632	1638	2169	2.3	0.3	1.1	1.6	1	0.2	1.6	0.2	0.4	RI, MS
**46**	**α-Cadinol**	1645	1645	2231	5.5	tr	tr	tr	1.2	1.7	1.4	1.2	1.3	RI, MS
47	Eudesma-4(15)-7-dien-1-β-ol	1663	1672	2199	3.0	tr	0.5	0.2	0.3	0.3	0.5	0.6	1.1	RI, MS
48	α-Bisabolol	1681	1667	2333	0.6	1.7	1.1	1.8	1	2.7	1.8	3.9	8	RI, MS
49	14-Hydroxy-α-muurolene	1755	1759	2599	0.5	1.6	0.4	0.5	0.3	0.5	0.5	0.5	0.9	RI, MS
50	14-Hydroxy-δ-cadinene	1788	1784	2607	1.0	0.2	0.1	0.6	0.4	0.6	0.4	0.6	0.3	RI, MS
51	Hexadecanoic acid	1942	1941	2930	1.2	0.2	0.5	0.4	0.1	0.8	0.1	0.9	0.4	RI, MS
52	Campestrolide	2142	2143	2970	10.3	50.9	32.1	16.3	8.1	9.5	14.6	16.3	4.5	RI, MS
Total Identification					73.1	84.4	81.6	91.8	84	85.1	84.9	78.6	78.0	
Yields						0.22	0.19	0.17	0.16	0.14	0.20	0.33	0.17	
Hydrocarbon compounds					33.4	20.3	35.9	50.5	59.2	63.8	58.4	44.9	50.2	
Oxygenated compounds					39.7	64.1	45.7	41.3	24.8	21.3	26.5	33.7	27.8	
Sesquiterpene compounds					60.5	31.7	45.3	69.6	65.9	70.9	68.3	59.8	67.9	
Oxygenated sesquiterpenes					27.4	12	12.5	24.2	15.7	10.6	11.8	16.3	22.6	
Hydrocarbon sesquiterpenes					33.1	19.7	32.8	45.4	50.2	60.3	56.5	43.5	45.3	
Monoterpene compounds					0.3	0.6	3.1	5.1	9.0	3.5	1.9	1.4	4.9	
Non terpenic compounds					12.3	52.1	33.2	17.1	9.1	10.7	14.7	17.4	5.2	

^a^ Order of elution is given on apolar column (Rtx-1). Bold types refer to main compounds. ^b^ Retention indices of literature on the apolar column (lRIa). ^c^ Retention indices on the apolar Rtx-1 column (RIa). ^d^ Retention indices on the polar Rtx-Wax column (RIp). ^e^ EO-CO: Standard sample of *E. campestre* essential oil. Quantification was carried out using Response Factors (RFs) relative to tridecane as internal standard. %: Normalized percentages are given on the apolar column except for components with identical RIa (percentages are given on the polar column). tr = trace (<0.05%). ^f^ Seasonal variation: Four sampling periods between May and July 2016. ^g^ Plant organs: Stems, leaves, flowers and root. ^h^ RI: Retention indices; MS: Mass spectrometry in electronic impact mode; Ref. comparison with literature data. All compounds were identified by comparing their EI-MS and retention indices with references compiled in the in-house library.

**Table 4 molecules-24-02575-t004:** Antimicrobial activities of *E. campestre* collective essential oil expressed by the diameter inhibition zones and MIC values (μg/mL).

	Diameters (mm)			MIC ^a^ (μg/mL)		
Strains	*EC-CO*	ATB ^b^: GENT	ATB ^b^: AmB	*EC-CO*	ATB ^b^: GENT	ATB ^b^: AmB
**Yeasts**						
*Candida albicans ATCC 10231*	6	ND	22	-	ND	1
*Candida albicans IP 444*	6	ND	20	-	ND	1
***Gram +***						
*Bacillus subtilis* ATCC 6633	6	20	ND	-	4	ND
*Bacillus cereus* ATCC 11778	23	19	ND	250	4	ND
*Staphylococcus aureus* ATCC 25923	35	21	ND	125	2	ND
*Staphylococcus aureus* ATCC 33862	21	20	ND	250	2	ND
*Staphylococcus aureus* ATCC 29213	20	20	ND	250	2	ND
*Enterococcus faecalis* ATCC 29212	35	10	ND	125	2	ND
*Listeria monocytogenes* ATCC 19115	6	19	ND	-	2	ND
***Gram −***						
*Pseudomonas aeruginosa* ATCC 27853	6	11	ND	-	4	ND
*Pseudomonas fluorescens* ATCC *13525*	6	11	ND	-	4	ND
*Salmonella enteritidis* ATCC 2453	6	20	ND	-	4	ND
*Escherichia coli* ATCC 25922	6	15	ND	-	4	ND
*Klebsiella pneumoniae* ATCC 70603	6	11	ND	-	4	ND

^a^ MIC: Minimum inhibitory concentration (given as μg/mL). ^b^ ATB: Antibiotics: GENT: Gentamicine (15 μg), AmB: Amphotericin B (100 μg). ND: not determined.

**Table 5 molecules-24-02575-t005:** Cytotoxicity (WI38 and J774), antitrypanosomal (Tbb), and antileishmanial (Lmm) activities expressed in the IC50 (Mean ± SD in μg/mL and μM for pure compounds from at least six values).

	Cytotoxicity		Antiparasitic Activity		Selectivity Index	
	IC_50_ ± SD in µg/mL (µM for Pure Compound)				IC_50_ WI38/IC_50_ Parasite	
	WI38	J774	Tbb	Lmm	Tbb	Lmm
**EC-CO**	>25	20.09 ± 2.00	0.86 ± 0.15	>25	>28.9	ND
**S31**	9.79 ± 1.95	8.29 ± 1.79	1.87 ± 0.73	7.40 ± 0.28	5.2	1.3
**EC1**	5.95 ± 0.87	6.70 ± 1.65	1.89 ± 0.69	3.90 ± 0.41	3.1	1.5
**ECC1**	24.74 ±0.05	12.23 ± 1.42	0.57 ± 0.06	14.74 ± 1.11	43.7	1.7
**campestrolide**	5.20 ± 0.24	4.84 ± 0.10	0.59 ± 0.08	3.43 ± 0.02	8.9	1.5
	(19.24 ± 0.87)	(17.89 ± 0.35)	(2.17 ± 0.28)	(12.67 ± 0.09)		
**Positive control**	0.036 ± 0.022 (0.103 ± 0.062) ^a^	0.007 ± 0.005 (0.021 ± 0.013) ^a^	0.031 ± 0.012 (0.022 ± 0.008) ^b^	0.057 ± 0.008 (0.097 ± 0.014) ^c^		

WI38: Non cancer human fibroblasts; J774: Cancerous macrophage-like murine cells; Tbb: *Trypanosoma brucei brucei* (bloodstream forms); Lmm: *Leishmania mexicana mexicana* promastigotes; Selectivity index calculated for antiparasitic activities compared to WI38 cytotoxicity. Positive control (reference drug): ^a^ camptothecin, ^b^ suramine, ^c^ pentamidine.

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
