# Peer review of "Essential Oil of Algerian Eryngium campestre: Chemical Variability and Evaluation of Biological Activities"

_molecules, 2019, doi:10.3390/molecules24142575_

Round 1
Reviewer 1 Report
The authors present a manuscript entitled “Essential oil of Algerian Eryngium campestre: Chemical variability and evaluation of biological activities”. The topic falls within the journal's scopes and, in my opinion it is very topical and the contribution proposed by the authors is certainly interesting. The experimental part is correctly described and the final data seems clearly highighted. In order to improve the quality of the manuscript, I suggest to make only some variations, as follows:
Ln 22: please, replace “prepared” with “extracted”
Table 2: please, define “tr”. It’s important to take in accout LOD and LOQ to consider the identification and quantification of an analyte.
All the References must be accorded to the guidelines of the journal
Author Response
Response to Reviewer 1 Comments
The authors present a manuscript entitled “Essential oil of Algerian Eryngium campestre: Chemical variability and evaluation of biological activities”. The topic falls within the journal's scopes and, in my opinion it is very topical and the contribution proposed by the authors is certainly interesting. The experimental part is correctly described and the final data seems clearly highighted. In order to improve the quality of the manuscript, I suggest to make only some variations, as follows:
Ln 22: please, replace “prepared” with “extracted” : the term "prepared" has been replaced by "extracted"
Table 2: please, define “tr”. It’s important to take in accout LOD and LOQ to consider the identification and quantification of an analyte.: "tr" was defined in the caption of table 2, all components detected by GC-FID with normalized abundance under 0.05% were considered at trace level. The determination of the LOD and LOQ is not possible because we not carried out a true quantification using authentic standard of all essential oil components but we used peak normalisation, including FID response factors relative to tridecane used as internal standard, and expressed as normalized % abundances.
All the References must be accorded to the guidelines of the journal: The references were checked according to the guidelines of the journal
Reviewer 2 Report
Article
Essential oil of Algerian Eryngium campestre: Chemical variability and evaluation of biological activities
A brief summary
The Authors determined the chemical composition of Algerian E. campestre essential oils using GC-FID and GC/MS, studied the biological activities of the essential oils against microorganisms involved in some infections and the cytotoxicity and antiparasitic activities.
Broad comments
The idea to investigate chemical compositions and variability depending on various factors, as soon as biological activities of less known plants is highly appreciate.
The Authors used proper analytical and statistical tools for study of chemical variability of E. campestre essential oils. The style of presentation is appropriate for scientific journal. In general the paper is properly constructed, however few completions are required.
Specific comments
Line numbers:
24 Do the percentages presented refer to the content of identified compounds in essential oils or numbers (or even areas) of acquired signals?
112 Is it possible to describe odour of the essential oil?
120, 128, 179, 309 In these lines “EC-CO” form was used, while in the table 2 “EC/CO”.
292 Why the fresh plant material have been used, instead of dried one? For many reasons a dried herb raw material is used in the industry. Is drying process strongly affected on chemical compositions of this raw material / essential oil?
293 and Table 2. The Authors showed the seasonal variation of each identified compound by comparison four sampling period between May and July. So, when were the samples of individual organs taken? Why samples from all 37 sites were harvested for so long time (May and July)? Is it possible to compare the data obtained from samples harvested at different times?
Author Response
Response to Reviewer 2 Comments
A brief summary
The Authors determined the chemical composition of Algerian E. campestre essential oils using GC-FID and GC/MS, studied the biological activities of the essential oils against microorganisms involved in some infections and the cytotoxicity and antiparasitic activities.
Broad comments
The idea to investigate chemical compositions and variability depending on various factors, as soon as biological activities of less known plants is highly appreciate.
The Authors used proper analytical and statistical tools for study of chemical variability of E. campestre essential oils. The style of presentation is appropriate for scientific journal. In general the paper is properly constructed, however few completions are required.
Specific comments
Line numbers:
24 Do the percentages presented refer to the content of identified compounds in essential oils or numbers (or even areas) of acquired signals? The percentages refer to the content of identified essential oil components
112 Is it possible to describe odour of the essential oil? Yes, we complet the manuscript indicating that the oil exhibited a musk odour
120, 128, 179, 309 In these lines “EC-CO” form was used, while in the table 2 “EC/CO”. the form was harmonized and the manuscript was corrected using EC-CO form to qualify the name of the collective essential oil of E campestre.
292 Why the fresh plant material have been used, instead of dried one? For many reasons a dried herb raw material is used in the industry. Is drying process strongly affected on chemical compositions of this raw material / essential oil?
The fresh plant material was used for pratical reasons, such the disponibilty of the clevenger apparatus in the Algerian laboratory. For complet, results not reported confirm that the essential oil composition prepared from dried plants was not different to those obtained from fresh plant material
293 and Table 2. The Authors showed the seasonal variation of each identified compound by comparison four sampling period between May and July. So, when were the samples of individual organs taken? Why samples from all 37 sites were harvested for so long time (May and July)? Is it possible to compare the data obtained from samples harvested at different times?
For the study of the oil compositions of separated organs, plant organs were collected when the plant is at the flowerinfg stage.
Concerning the study of the chemical varaibility of the essential oil, the 37 plant samples were harvested in the same vegetative stage, as during the flowering stage. This information was added to the manuscript (see line 188). So the comparison of the oils is possible because all studied samples provinding to the same vegetative state.